# Stabilization of the Metastable Pre-Fusion Conformation of the SARS-CoV-2 Spike Glycoprotein through N-Linked Glycosylation of the S2 Subunit

**DOI:** 10.3390/v16020223

**Published:** 2024-01-31

**Authors:** Fuwen Zan, Yao Zhou, Ting Chen, Yahan Chen, Zhixia Mu, Zhaohui Qian, Xiuyuan Ou

**Affiliations:** 1NHC Key Laboratory of Systems Biology of Pathogens, National Institute of Pathogen Biology, Chinese Academy of Medical Sciences & Peking Union Medical College, Beijing 102629, Chinazhouyao@ipbcams.ac.cn (Y.Z.);; 2Key Laboratory of Pathogen Infection Prevention and Control (Ministry of Education), National Institute of Pathogen Biology, Chinese Academy of Medical Sciences & Peking Union Medical College, Beijing 102629, China; 3State Key Laboratory of Respiratory Health and Multimorbidity, National Institute of Pathogen Biology, Chinese Academy of Medical Sciences & Peking Union Medical College, Beijing 102629, China

**Keywords:** SARS-CoV-2, S protein, N-linked glycosylation, syncytia, stability

## Abstract

Severe acute respiratory syndrome coronavirus 2 (SARS-CoV-2), the novel coronavirus responsible for the coronavirus disease 2019 (COVID-19) pandemic, represents a serious threat to public health. The spike (S) glycoprotein of SARS-CoV-2 mediates viral entry into host cells and is heavily glycosylated. In this study, we systemically analyzed the roles of 22 putative N-linked glycans in SARS-CoV-2 S protein expression, membrane fusion, viral entry, and stability. Using the α-glycosidase inhibitors castanospermine and NB-DNJ, we confirmed that disruption of N-linked glycosylation blocked the maturation of the S protein, leading to the impairment of S protein-mediated membrane fusion. Single-amino-acid substitution of each of the 22 N-linked glycosylation sites with glutamine revealed that 9 out of the 22 N-linked glycosylation sites were critical for S protein folding and maturation. Thus, substitution at these sites resulted in reduced S protein-mediated cell–cell fusion and viral entry. Notably, the N1074Q mutation markedly affected S protein stability and induced significant receptor-independent syncytium (RIS) formation in HEK293T/hACE2-KO cells. Additionally, the removal of the furin cleavage site partially compensated for the instability induced by the N1074Q mutation. Although the corresponding mutation in the SARS-CoV S protein (N1056Q) did not induce RIS in HEK293T cells, the N669Q and N1080Q mutants exhibited increased fusogenic activity and did induce syncytium formation in HEK293T cells. Therefore, N-glycans on the SARS-CoV and SARS-CoV-2 S2 subunits are highly important for maintaining the pre-fusion state of the S protein. This study revealed the critical roles of N-glycans in S protein maturation and stability, information that has implications for the design of vaccines and antiviral strategies.

## 1. Introduction

Severe acute respiratory syndrome coronavirus 2 (SARS-CoV-2), the causative agent of coronavirus disease 2019 (COVID-19), poses a significant threat to global public health [1,2]. As of 22 November 2023, there have been more than 772 million confirmed cases worldwide, resulting in more than 6.97 million deaths (https://covid19.who.int/?mapFilter=cases, accessed on 22 November 2023). SARS-CoV-2 is an enveloped, single-strand, positive-sense RNA virus that belongs to the *Sarbecovirus* lineage of the *Betacoronavirus* genus in the *Coronaviridae* family, which also includes the highly pathogenic virus SARS-CoV [3]. Although there are approved vaccines for SARS-CoV-2, the virus is continuously evolving, resulting in the emergence of new variants, such as the Omicron variants. These new variants show markedly increased immune evasion, exhibiting decreasing vulnerability to neutralization by COVID-19 convalescent and vaccinated sera. These developments indicate that novel vaccine designs are urgently needed [4]. A better understanding of the structure and function of viral glycoproteins would aid in the development of novel antiviral therapeutic targets.

The coronavirus spike (S) glycoproteins form homotrimeric spikes on the virion surface and mediate viral entry into host cells [5]. The monomeric S protein consists of two subunits, S1 and S2, which are responsible for receptor binding and membrane fusion, respectively. In SARS-CoV-2, the S protein is cleaved into S1 and S2 by furin in the Golgi apparatus during S protein maturation [4]. The native trimeric S protein is metastable; upon receptor binding and proteolytic cleavage, it undergoes a cascade of conformational changes from an initial metastable pre-fusion conformation to a stable post-fusion conformation, in which it drives the fusion of viral and cellular membranes [6,7].

Like those of the human immunodeficiency virus (HIV) envelope protein, influenza virus hemagglutinin, and Ebola virus glycoprotein, the S proteins of coronaviruses are heavily glycosylated [8,9]. The trimeric S proteins of SARS-CoV, SARS-CoV-2, and MERS-CoV contain 69, 66, and 69 putative N-linked glycosylation sites, respectively [8,9]. The S protein is synthesized in the endoplasmic reticulum (ER) and transported to the Golgi apparatus for further modifications. N-linked glycosylation is a common post-translational modification in enveloped viruses, and the asparagine residue of the N-linked glycosylation sequon (Asn-X-Thr/Ser, where X is not Pro) is glycosylated [10]. Mass spectrometry (MS) analysis revealed extensive heterogeneity in the N-glycans of the S protein, which include high-mannose-type glycans, hybrid-type glycans and complex-type glycans [9,11]. N-glycans cover approximately three quarters of the surface of the trimeric S protein, shielding epitopes on the S protein from the host immune response and attenuating viral immunogenicity [8,12]. In parallel with epitope masking, glycan-dependent epitopes of viral glycoproteins can induce the production of neutralizing antibodies [13]. The N-glycans of the SARS-CoV S protein are also involved in viral attachment to attachment factors, such as DC/L-SIGN, and the N-glycosites N227 and N699 on the SARS-CoV S protein are critical for binding to DC/L-SIGN [14]. It has been reported that N-glycans at positions N165, N234, and N343 of the SARS-CoV-2 S protein stabilize the RBD in an open state and thus contribute to an increase in infectivity [15,16,17], whereas N-glycans at position N370 stabilize the RBD in a closed state, resulting in decreased infectivity [18,19]. A recent study revealed that N-glycans on the stalk and hinge regions of the HCoV-NL63 S protein modulate protein bending [20]. Although SARS-CoV-2 evolves continuously and new variants emerge and disappear, most S protein N-linked glycosylation sites are highly conserved among variants, except for the deletion of the N17 glycosite in the Delta variant and the introduction of the N20 and N188 glycosites in the Gamma variant. This overall conservation indicates the importance of N-glycosylation in S proteins [21]. In the present study, we investigated the roles of N-glycans on the SARS-CoV-2 S glycoprotein in protein expression, stability, membrane fusion, and viral entry, and found that some N-glycans on the SARS-CoV and SARS-CoV-2 S2 subunits are important for maintaining the structure of the S protein in the pre-fusion state.

## 2. Materials and Methods

### 2.1. Plasmid Construction

A codon-optimized SARS-CoV-2 S gene (GenBank: QHU36824.1) with a deletion of the last 19 amino acids was synthesized with GenScript (Nanjing, China) and cloned into pcDNA3.1(+) between the *Hind* III and *BamH* I sites. Plasmids encoding the SARS-CoV S protein with a deletion of the last 19 amino acids have been previously described [22]. All mutagenesis experiments were performed using a Q5 mutagenesis kit (NEB, Ipswich, MA, USA). After the entire coding sequence was verified, the coding regions were subcloned into pcDNA3.1(+). Human ACE2-targeting gRNA (5′-GAAAGCTGGAGATCTGAGGT-3′) was synthesized and cloned into the pLentiCRISPRv2 vector to generate the recombinant plasmid pCRISPRv2-hACE2.

### 2.2. Cell Lines and Culture Conditions

Human embryonic kidney (HEK) 293T cells, HEK293 cells, and HEK293 cells stably expressing human angiotensin-converting enzyme 2 (293/hACE2) were cultured in Dulbecco’s modified Eagle’s medium (DMEM) supplemented with 10% fetal bovine serum (FBS) and 1% penicillin–streptomycin–fungizone (PSF) at 37 °C with 5% CO_2_. The endogenous ACE2 knockout HEK293T cell line (HEK293T/hACE2-KO) was generated using the CRISPR–Cas9 system.

### 2.3. Production and Transduction of S-Pseudotyped Lentiviruses

Plasmids encoding different S coronavirus proteins were co-transfected with psPAX2 and pLenti-Luc-GFP into HEK293T cells. After 40 h of incubation, the pseudovirion-containing supernatants were centrifuged at 1000× *g* for 10 min to remove cell debris. To quantify the transduction efficiency of the S-pseudotyped lentiviruses, HEK293/hACE2 cells were seeded in a 24-well plate, and after overnight incubation, the cells were inoculated with pseudovirions. At 40 h post-inoculation, cells were lysed using Steady-Glo Reagent (Promega, Madison, WI, USA). Transduction efficiency was determined via a quantifying luciferase activity using a PerkinElmer EnVision^®^ Multilabel Plate Reader.

### 2.4. Detection of Viral Spike Glycoproteins Using Western Blotting

To evaluate S protein expression in the cell lysates, HEK293T cells were transfected with plasmids encoding different S proteins. At 40 h post-transfection, the cells were lysed using RIPA lysis buffer containing protease inhibitors. To determine the incorporation of S proteins into the pseudovirions, the pseudovirus-containing supernatant was pelleted through a 20% sucrose cushion at 25,000 rpm at 4 °C for 2 h. Viral pellets were resuspended in 1 × loading buffer. Cell lysates and pseudovirion pellets were separated using 10% sodium dodecyl–sulfate polyacrylamide gel electrophoresis (SDS-PAGE) and transferred to nitrocellulose membranes. The SARS-CoV-2 S proteins were detected with a rabbit polyclonal anti-S2 antibody (Sinobiological Inc., Beijing, China), and the blot was further stained with horseradish peroxidase-conjugated goat anti-rabbit IgG and visualized with Clarity Western ECL substrate (Bio-Rad, Hercules, CA, USA). β-actin and HIV capsid protein (p24) were detected using a mouse monoclonal anti-β-actin antibody (Sigma, St. Louis, MO, USA) and a rabbit polyclonal anti-p24 antibody (Sinobiological Inc., Beijing, China), respectively.

### 2.5. Flow Cytometric Analysis of Cell Surface S Protein Expression

HEK293T cells transiently expressing different S proteins were digested with 1 mM ethylenediaminetetraacetic acid (EDTA). After washing, the cells were incubated with polyclonal rabbit anti-SARS-CoV-2 S2 antibodies (Sinobiological Inc., Beijing, China) for 1 h on ice. This step was followed by incubation with FITC-conjugated goat anti-rabbit IgG (Jackson ImmunoResearch, West Grove, PA, USA) for 1 h. After washing, the cells were analyzed using a FACSCalibur flow cytometer (Becton Dickinson, Franklin Lakes, NJ, USA).

### 2.6. Soluble hACE2 Binding Assay

HEK293T cells transiently expressing different S proteins were digested with 1 mM EDTA. After washing, the cells were incubated with 5 μg/mL soluble hACE2 for 1 h on ice. Subsequently, the cells were washed and incubated with 1 µg/mL polyclonal goat anti-human ACE2 antibody (R&D Systems, Minneapolis, MN, USA) for 1 h. Next, the cells were incubated with FITC-conjugated rabbit anti-goat secondary antibody (Jackson ImmunoResearch, West Grove, PA, USA) for 1 h. After washing, the cells were analyzed using a FACSCalibur flow cytometer (Becton Dickinson, Franklin Lakes, NJ, USA).

### 2.7. Cell–Cell Fusion Assay

HEK293T cells were co-transfected with plasmids encoding different S proteins and GFP in the absence or presence of 100 μg/mL castanospermine (#HY-N2022, MCE) or NB-DNJ (#HY-17020, MCE). After 40 h of transfection, the cells were detached using 0.25% trypsin for 2 min and approximately 1/3 of these cells were overlaid on HEK293/hACE2 cells. After 4 h of incubation, images of syncytia were captured using a Nikon TE2000 epifluorescence microscope with MetaMorph software version 7.5 (Molecular Devices, Sunnyvale, CA, USA).

### 2.8. Detection of S1 Subunit Shedding

HEK293T cells were transfected with plasmids encoding either wild-type or mutant S protein. After 40 h, the cell culture media were collected and centrifuged at 1000× *g* for 10 min to remove cell debris. The supernatants were then concentrated using Amicon Ultra-0.5 mL 3 kDa Centrifugal Filters (Millipore, Cork, Ireland). Thereafter, the S1 subunits in the concentrated supernatants were detected by Western blot with antibodies against the SARS-CoV-2 RBD.

### 2.9. Cell Counting Kit-8 (CCK-8) Assay

To estimate the viability of the HEK293T cells in vitro, a CCK-8 assay was conducted according to the manufacturer’s instructions (Vazyme, Nanjing, China). HEK293T cells were seeded into a 96-well plate and cultured in 100 µL of DMEM supplemented with 10% FBS. After overnight attachment, the seeded cells were treated with 100 µg/mL castanospermine or NB-DNJ for 40 h. Afterwards, 10 µL CCK-8 reagent was added to each well and the cells were incubated for 2 h. The absorbance of each well was measured at a wavelength of 450 nm using an automicroplate reader.

### 2.10. Statistical Analysis

The data are presented as the mean ± standard deviation of three experimental replicates, as determined using GraphPad Prism version 7. Comparisons between means were analyzed using the unpaired two-tailed Student’s *t*-test.

## 3. Results

### 3.1. α-Glucosidase Inhibitors Reduced the Efficiency of SARS-CoV-2 S-Mediated Membrane Fusion

To determine the effect of N-glycans on SARS-CoV-2 S protein expression, HEK293T cells transiently expressing the SARS-CoV-2 S protein were treated with the α-glycosidase inhibitors castanospermine and NB-DNJ and the levels of S protein in the cell lysates were analyzed via Western blot. Consistent with our previous reports [23], both cleaved and full-length (FL), uncleaved forms of the S protein were detected using antibodies against SARS-CoV-2 S2 (Figure 1A). There were two close-together bands corresponding to the FL S proteins. The higher, fainter band may represent the fully glycosylated FL S proteins that are cleaved to generate S1 and S2 in the Golgi apparatus and plasma membrane during S protein maturation, whereas the lower FL band likely represents nascent, unglycosylated FL S proteins. Treatment with α-glycosidase inhibitors had almost no effect on the levels of FL S proteins, but the levels of cleaved S proteins in the α-glycosidase inhibitor-treated group were significantly lower than those in the dimethyl sulfoxide (DMSO)-treated group (Figure 1A), indicating that inhibition of N-linked glycosylation might strongly affect S protein folding and maturation. Previously, we and others showed that co-culturing SARS-CoV-2 S protein-expressing HEK293T cells with HEK293/hACE2 cells in the presence of trypsin induced significant cell–cell fusion or syncytium formation. As shown in Figure 1B, significant syncytium formation was observed when WT SARS-CoV-2 S-expressing cells were mixed with HEK293/hACE2 cells in the presence of trypsin. However, syncytium formation was significantly reduced, as shown in Figure 1B, when SARS-CoV-2 S protein-expressing HEK293T cells were pretreated with castanospermine or NB-DNJ, a result consistent with the lower levels of S protein expression observed in cells treated with α-glycosidase inhibitors (Figure 1A). Neither castanospermine nor NB-DNJ has any inhibitory effects on the viability of HEK293T cells at the concentrations used in our study, as determined through the CCK8 assay (Figure 1C). We also evaluated the effects of glycosidase inhibitors on S-mediated viral entry by SARS-CoV-2 S-pseudotyped lentiviruses. Treatment of cells with α-glycosidase inhibitors also greatly decreased the amount of SARS-CoV-2 S protein incorporated into the pseudovirions, resulting in a significant reduction in the transduction efficiency of SARS-CoV-2 S pseudovirions with respect to HEK293/hACE2 cells (Figure 1D,E). Together, these results indicate that the inhibition of complex-type glycans in the SARS-CoV-2 S protein by the α-glycosidase inhibitors castanospermine and NB-DNJ strongly affects S protein maturation, thus impairing S-mediated membrane fusion and virus entry.

### 3.2. Effect of Individual N-Linked Glycosylation on the Levels of Expression, Surface Presentation, and Receptor Binding of SARS-CoV-2 S Proteins

The SARS-CoV-2 S protein contains 22 putative N-linked glycosylation sites, 13 in the S1 subunit and 9 in the S2 subunit (Figure 2A). We determined which glycosylation site(s) might affect S protein folding and maturation. The asparagine (N) residue in each glycosylation site (NxT/S) was substituted with glutamine (Q) to remove the N-linked glycosylation. Individual single mutations had almost no effect on the expression levels of the FL S proteins, whereas 9 out of 22 mutants showed a marked decrease in the levels of cleaved S proteins compared to cells expressing the WT S protein (Figure 2B). The N61Q, N343Q, and N801Q mutants resulted in levels of cleaved S proteins only slightly above the background levels, and the N122Q, N165Q, N331Q, N717Q, N1098Q, and N1173Q mutations also resulted in a marked reduction in levels of cleaved S proteins, indicating that N-linked glycosylation might have a significant effect on S protein folding and maturation.

Next, we evaluated the levels of individual mutant S proteins on the cell surface and their ability to bind to their cognate receptor, hACE2. This value was determined via flow cytometry. As shown in Figure 2C,D, the levels of each mutant S protein on the cell surface and the associated levels of receptor binding were largely consistent with the amounts of the cleaved S proteins in the cell lysate.

### 3.3. Effect of Individual N-Linked Glycans on SARS-CoV-2 S-Mediated Cell–Cell Fusion and Pseudovirus Entry

To investigate whether each individual N-glycosite affects S protein-mediated membrane fusion, we measured the levels of mutant S protein-mediated cell–cell fusion. Most of the mutants induced cell–cell fusion at levels comparable with the amount of fusion induced by the wild-type S protein. However, N61Q-, N122Q-, N165Q-, N331Q-, N343Q-, N801Q-, and N1074Q-mediated cell–cell fusion was significantly impaired, a result consistent with the markedly lower expression of these mutant S proteins on the cell surface (Figure 3A,B).

### 3.4. Effect of SARS-CoV-2 S Protein N-Glycans on S Protein-Mediated Viral Entry

To further characterize the role of individual N-glycosites of the SARS-CoV-2 S protein in S-mediated membrane fusion, we used a lentiviral pseudotype system to evaluate mutant S protein-mediated viral entry. The majority of the mutant S proteins were associated with a reduction in transduction efficiency to various degrees (Figure 4A). Compared to that of the WT S protein, the transduction efficiencies of the N61Q, N122Q, N343Q, and N801Q mutants were reduced more than 10-fold, whereas the N165Q, N331Q, N657Q, N717Q, and N1098Q pseudovirus mutants showed more than 5-fold but less than 10-fold reductions in infectivity. Overall, the transduction efficiency of the mutant S proteins largely correlates with the levels of S protein incorporation into the pseudovirions (Figure 4B). The N61Q, N122Q, N165Q, N331Q, N343Q, N717Q, N801Q, N1098Q, and N1173Q mutants showed a marked reduction in S protein incorporation into pseudovirions to various degrees (Figure 4B), a finding consistent with their relatively low expression levels in cell lysates (Figure 2B).

### 3.5. Removal of N-Glycans in N1074 Reduced the Stability of SARS-CoV-2 S Protein and Induced Receptor-Independent Cell–Cell Fusion in HEK293T Cells

Surprisingly, when HEK293T cells were transfected with N1074Q, a significant number of syncytia were observed, even without the expression of exogenous ACE2 (Figure 5A). To further determine whether the removal of N-glycans at N1074 was responsible for syncytium formation in HEK293T cells, we also substituted the threonine residue at position 1076 with an alanine residue to abolish the N-linked glycosylation sequon around N1074. The T1076A mutant showed a phenotype similar to that of the N1074Q mutant, which further confirmed that the loss of N-linked glycosylation at position 1074 on the SARS-CoV-2 S protein caused syncytium formation in HEK293T cells (Figure 5A). To investigate whether N1074Q- and T1076A-induced syncytium formation in HEK293T cells is ACE2-dependent, the formation of syncytia in the presence of these two mutants was also evaluated in endogenous-ACE2-knockout HEK293T cells. We knocked out endogenous ACE2 in HEK293T cells using the CRISPR–Cas9 system, and the knockout was confirmed through sequencing (Figure 5B). Next, the N1074Q and T1076A mutants were transiently expressed in endogenous ACE2 knockout HEK293T cells; both mutants also induced syncytium formation in endogenous-ACE2-knockout HEK293T cells. Thus, we conclude that the removal of N-glycans at N1074 of SARS-CoV-2 induced receptor-independent syncytium formation in HEK293T cells.

We hypothesized that the N1074Q and T1076A mutations might weaken the association between the S1 subunit and the rest of the S protein, resulting in an increase in S1 shedding and spontaneous conformational changes, which culminate in syncytium formation. To test our hypothesis, we determined the amount of S1 in the supernatants of HEK293T cells transiently expressing either the wild-type or a mutant S protein. Consistent with our hypothesis, both the N1074Q and the T1076A mutants were associated with more S1 shedding than the WT S protein was (Figure 5C). Collectively, these data indicate that the glycans in N1074 on the SARS-CoV-2 S protein are important for maintaining the stability of the S protein, shedding of the S1 subunit, and triggering cell–cell fusion in HEK293T cells.

### 3.6. Removal of the Furin Cleavage Site from the SARS-CoV-2 S Protein Compensated for the Instability of the N1074Q Mutant

The SARS-CoV-2 S protein contains a furin cleavage site at the S1/S2 boundary, which facilitates the entry of SARS-CoV-2 into human respiratory cells. The removal of the furin cleavage site significantly increases the overall stability of the S protein [24,25,26]. Next, we determined whether removal of the furin cleavage site might stabilize the N1074Q S protein. Consistent with previous reports [22], deletion of the furin cleavage site reduced cleavage of the S protein, as evidenced by the absence of the cleaved form of the S protein in the cells expressing WT S without a furin cleavage site (WT-delFurin) and cells expressing the N1074Q mutant without a furin cleavage site (N1074Q-delFurin) (Figure 6A). As shown in Figure 6B, the removal of the furin cleavage site significantly stabilized the S protein, and the N1074Q-delFurin mutant induced only minimal syncytium formation in HEK293T cells, while the N1074Q mutant resulted in extensive syncytium formation. As a control, a receptor-dependent cell–cell fusion assay was also performed. To prevent interference from syncytium formation in HEK293T cells expressing mutant S proteins in a receptor-dependent cell–cell fusion assay, we performed the assay at 30 h post-transfection. At that time, no obvious syncytia were observed in the HEK293T cells expressing the N1074 mutant. We observed that all mutant S proteins, compared to WT S proteins, induced similar levels of syncytium formation in the presence of trypsin, indicating that removal of the furin cleavage site has almost no effect on the overall function of the S protein (Figure 6C).

### 3.7. N-Glycosylation of the S2 Subunit Contributes to Stability of the SARS-CoV S Protein

There are nine N-linked glycosylation sites in the SARS-CoV-2 S2 subunit that are also conserved in the SARS-CoV S2 (Figure 7A). We decided to determine whether the N-glycans in the SARS-CoV S2 subunit have similar functions. We mutated each of the nine N-linked glycosylation sites in the SARS-CoV S2 subunit and analyzed the expression levels of the mutant S proteins via Western blot. All mutants were expressed, as was the WT S protein (Figure 7B). Strikingly, the N1056Q mutation in the SARS-CoV S protein, corresponding to the N1074Q mutation in the SARS-CoV-2 S protein, did not induce marked syncytium formation in HEK293T cells (Figure 7C). In contrast, the N699Q and N1080Q mutants triggered extensive cell–cell fusion. The N1176Q mutant also induced low-level syncytium formation in HEK293T cells at 40 h post-transfection. We further investigated whether N699Q- and N1080Q-induced syncytium formation were ACE2-dependent. Notably, the levels of syncytium formation induced by both mutants in HEK293T/hACE2-KO cells were comparable to those induced in HEK293T cells (Figure 7D). Together, these results indicate that the N-glycans on the S2 subunit are also involved in the stability of the SARS-CoV S protein.

## 4. Discussion

N-linked glycosylation is a glycosylation modification commonly found in enveloped viruses that contributes to protein structure and function. Like other type-I viral fusion proteins, the SARS-CoV-2 S protein is heavily glycosylated [8,9]. In this study, we showed that N-linked glycosylation at residue N1074 of the SARS-CoV-2 S2 subunit is crucial for stability of the S protein. The removal of the N-glycans from position N1074 in the SARS-CoV-2 S protein via mutation at either N1074Q or T1076A triggered RIS formation in HEK293T/hACE2-KO cells. We previously reported that the RIS of the S protein of mouse hepatitis virus (MHV) A59 is critical for virus fitness [27]. However, MHV-A59 S-induced RIS was pH-dependent [27], whereas the SARS-CoV-2 N1074Q or T1076A mutant S protein-mediated RIS was not pH-dependent.

The SARS-CoV-2 D614G variant is an early human-adapted SARS-CoV-2 variant that rapidly became the dominant strain worldwide. The D614G mutation in the S protein stabilized the S1/S2 complex and inhibited S1 shedding, suggesting that the modulation of the S protein stability is critical in SARS-CoV-2 human adaptation [27,28]. N-glycans of viral fusion glycoproteins play a critical role in maintaining the conformation state of viral glycoproteins, therefore affecting viral infectivity and pathogenicity [29,30]. The avian influenza virus H5N6 increases its adaptation to mammalian hosts by acquiring an N-linked glycoprotein at the head of the HA protein that increases the stability of the HA trimeric structure [31]. N-glycans at residues N165 and N234 of the SARS-CoV-2 S protein modulate the conformational dynamics of the SARS-CoV-2 S protein by stabilizing the RBD in the “up” conformation [15,16]. Here, we found that the glycosylation sites at position N1074 on the SARS-CoV-2 S protein and at positions N699 and N1080 on the SARS-CoV S protein are highly important for maintaining the stability of the pre-fusion S trimers. Glycosylation site N1074 in the SARS-CoV-2 S protein is located in the connector domain between heptad repeat 2 and the central helix [9,32]. In the pre-fusion S trimer, residue N1074 is located at the bottom of a groove formed by two adjacent S monomers (Figure 8A). It is possible that the bulky hydrophilic carbohydrate chains at position N1074 interact with the side chains of the surrounding residues to fill the groove and aid in stabilizing the S trimer. Although the N-linked glycosylation sites of the S2 subunit are conserved between SARS-CoV-2 and SARS-CoV (Figure 8B), the corresponding mutant in the SARS-CoV S protein, the N1056Q mutant, showed no obvious difference in the stability of the S trimer. However, the N-glycans of two spatially close glycosylation sites, N699 and N1080, are vital for the stability of the SARS-CoV S trimer. The glycosylation sites N699 and N1080 are structurally adjacent to N1058 on the SARS-CoV S protein (Figure 8B). Therefore, oligosaccharides at these glycosylation sites may function together to maintain the stability of the pre-fusion S trimer, although some sites may play a more important role than others. Further functional and structural analyses are required to confirm this hypothesis.

The pre-fusion state of the trimeric S protein is metastable, and the protein transforms into a stable post-fusion state upon receptor binding and protease cleavage or antibody binding while mediating membrane fusion [3,6]. The presence of the furin cleavage site in the S protein of SARS-CoV-2 facilitates both protease cleavage around the S2′ site and conformational changes in the S trimer during membrane fusion, but it does so at the cost of protein stability [25]. Although the deletion of the furin cleavage site in N1074Q mutant did not affect S-mediated receptor-dependent membrane fusion, it no longer triggered cell–cell fusion in HEK293T cells, indicating that elimination of the furin cleavage site from the S protein compensates for the instability of the N1074Q mutant. Unlike the SARS-CoV-2 S protein, the S protein of SARS-CoV does not have a furin cleavage site at the boundary of the S1 and S2 subunits, but the removal of N-glycans at position N699 or N1080 of the SARS-CoV S protein still induces RIS formation in HEK293T cells, indicating subtle differences between the two S proteins despite the high similarity of the S2 subunits.

The N-linked glycosylation of nascent polypeptides is initiated in the ER cotranslationally. This process is thought to be critical for proper folding. Further modifications to glycans occur in the Golgi network [33]. N-glycans at N61 and N801, which are structurally close to the S1/S2 and S2′ proteolytic sites, respectively, in the SARS-CoV-2 S trimeric structure have been shown to regulate S protein maturation and viral infectivity [34]. Deletion of N122 glycosylation in the NTD of the SARS-CoV-2 S protein resulted in reduced infectivity and low protein expression [35]. Here, we found that the removal of 9 out of the 22 N-linked glycosylation sites (N61, N122, N165, N331, N343, N717, N801Q, N1098, and N1173) impaired the cleavage of the S protein, a result that may be due to improper protein folding and impaired trafficking. This change reduced S-mediated membrane fusion. This finding is in contrast to findings for the SARS-CoV S protein, for which no individual mutation of any of the putative N-linked glycosylation sites on the S1 subunit had a major effect on its expression [14]. The ablation of glycosylation at both N331 and N343 in the RBD of the SARS-CoV-2 S protein significantly reduced infectivity [13]. This finding is further supported by our findings that the N-glycans at N331 and N343 affect protein maturation (Figure 2B). N165 has been reported to contain mainly processed glycans that reside at the crevice between the NTD from one chain and the RBD from an adjacent chain. The removal of N-glycans from N165 of the SARS-CoV-2 S protein may cause improper S trimer folding or stability, ultimately impairing the function of the mutant S protein. The N717, N1098, and N1173 glycosylation sites of the SARS-CoV-2 S protein are located close to the base of the intact spike protein, but the mechanism of their participation in protein maturation remains to be determined. A recent study showed that N-linked glycosylation has a lesser effect on S protein expression in the SARS-CoV-2 Omicron variants compared to its effect on the D614G S protein [36].

As a major surface protein, the coronavirus S protein is the focus of vaccine development. The ectodomain S trimer from MERS-CoV has been shown to elicit more potent neutralizing antibodies than the S1 monomer or the RBD [37]. However, the pre-fusion state of the S trimer from SARS-CoV-2 is unstable, as evidenced by the presence of the post-fusion state in SARS-CoV-2 intact virions and β-propiolactone-inactivated SARS-CoV-2 virions [5,38]. Cryo-EM structures of coronavirus S glycoproteins have revealed great differences between the pre- and post-fusion states, indicating that post-fusion S immunogens may not efficiently induce neutralizing antibodies [39]. Therefore, many studies have been conducted to stabilize the pre-fusion SARS-CoV-2 S trimer, with approaches that include the removal of the furin cleavage site, the introduction of two consecutive proline substitutions (K986P and V987P) to the turn between heptad repeat 1 and the central helix, and the introduction of six proline substitutions (F817P, A892P, A899P, A942P, K986P, and V987P) in the S2 subunit [25,40,41]. Our study demonstrates that N-glycosylation in the SARS-CoV-2 S2 subunit is involved in maintaining the stability of the S trimer. The introduction of N-glycosylation to the SARS-CoV-2 S2 subunit may be a novel approach by which to improve the stability of the pre-fusion SARS-CoV-2 S trimer, but this finding needs to be confirmed by further studies.

## 5. Conclusions

Collectively, these findings demonstrate that the N-linked glycosylation of the SARS-CoV-2 S protein is highly important for protein maturation and for ensuring the stability of the pre-fusion state. These data provide insights into the complex pathway involved in the biosynthesis of the SARS-CoV-2 S glycoprotein and have valuable implications for the development of recombinant vaccines and antiviral therapeutics.

## Figures and Tables

**Figure 1 viruses-16-00223-f001:**
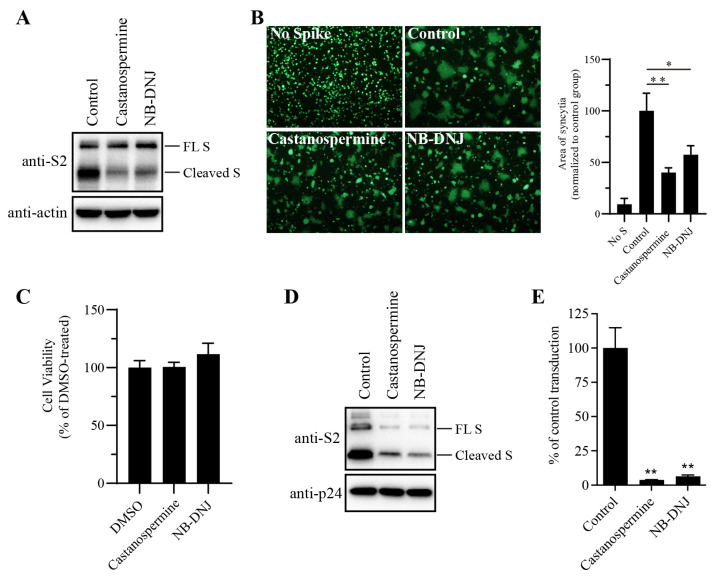
α-glycosidase inhibitors impaired SARS-CoV-2 S-mediated membrane fusion. (**A**) HEK293T cells transfected with plasmids expressing the SARS-CoV-2 S protein were treated with 100 µg/mL castanospermine or NB-DNJ for 40 h, after which the expression levels of S protein in cell lysates were detected using Western blot analysis with antibodies against the S2 subunit of the SARS-CoV-2 S protein. β-actin was used as a loading control. (**B**) HEK293T cells expressing the SARS-CoV-2 S protein and GFP were treated with α-glycosidase inhibitors for 40 h and then trypsinized at 40 h post-transfection and co-cultured with HEK293/hACE2 for another 4 h, then imaged using a fluorescence microscope. The area of syncytia in each image was measured using ImageJ. An unpaired two-tailed Student’s *t*-test was used for statistical analysis, ** *p* < 0.01; * *p* < 0.05. (**C**) Effect of α-glycosidase inhibitors on the proliferation of HEK293T cells. HEK293T cells were treated with 100 µg/mL castanospermine or NB-DNJ for 40 h, after which cell viability was determined using the CCK8 assay. (**D**) Western blot analysis of SARS-CoV-2 S-pseudotyped lentiviral particles produced in HEK293T cells in the absence or presence of α-glycosidase inhibitors. p24 was used as a loading control. (**E**) The transduction efficiency of SARS-CoV-2 S-pseudotyped lentiviral particles produced in HEK293T cells in the absence or presence of α-glycosidase inhibitors. An unpaired two-tailed Student’s *t*-test was used for statistical analysis, ** *p* < 0.01.

**Figure 2 viruses-16-00223-f002:**
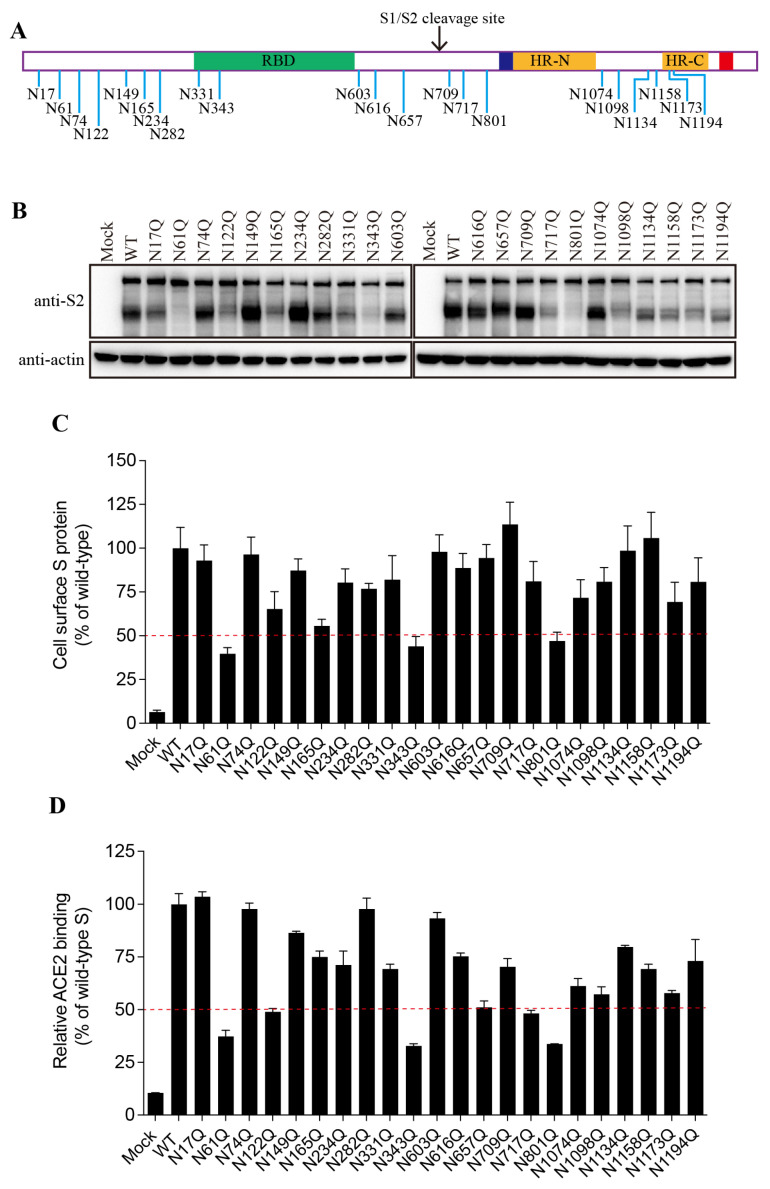
The expression levels and receptor-binding capacity of the N-glycosylation-site mutants of the SARS-CoV-2 S protein. (**A**) Schematic representation of N-linked glycosylation sites in the SARS-CoV-2 S protein. (**B**) HEK293T cells were transfected with plasmids encoding wild-type S protein or with N-glycosylation-site mutants. After 40 h of transfection, the cells were lysed and analyzed via Western blot with antibodies against the SARS-CoV-2 S2 subunit. β-actin was used as a loading control. (**C**) Cell-surface expression of wild-type S protein and different N-glycosylation-site mutants, determined via flow cytometry. (**D**) HEK293T cells were transfected with plasmids encoding wild-type S protein or different N-glycosylation mutants. After 40 h of transfection, the cells were detached with 1 mM EDTA, incubated with soluble hACE2 for 1 h on ice, incubated with primary antibodies against hACE2, and then incubated with FITC-conjugated secondary antibodies. The cells were then subjected to flow cytometry.

**Figure 3 viruses-16-00223-f003:**
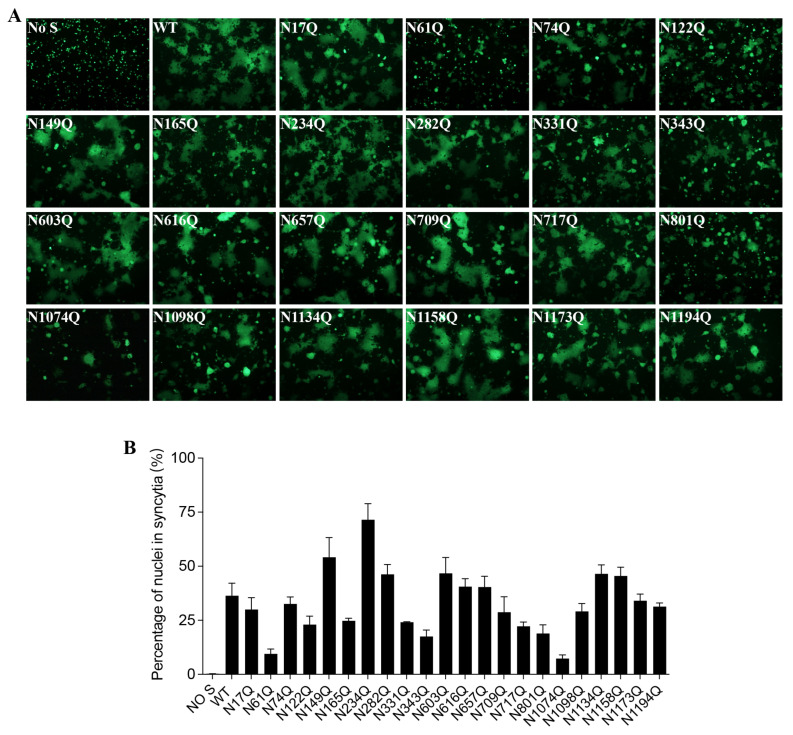
Cell–cell fusion capacity of N-glycosylation mutants of the SARS-CoV-2 S protein. (**A**) HEK293T cells were co-transfected with plasmids encoding either wild-type S protein or different N-glycosylation-site-mutant S proteins and plasmids encoding GFP. After 40 h of transfection, the cells were trypsinized and co-cultured with HEK293/hACE2 cells for an additional 4 h and then imaged using a fluorescence microscope. (**B**) Quantification of cell–cell fusion. The total number of nuclei and the number of nuclei in fused cells for each image were counted. The fusion efficiency was calculated as the number of nuclei in syncytia/the total number of nuclei ×100.

**Figure 4 viruses-16-00223-f004:**
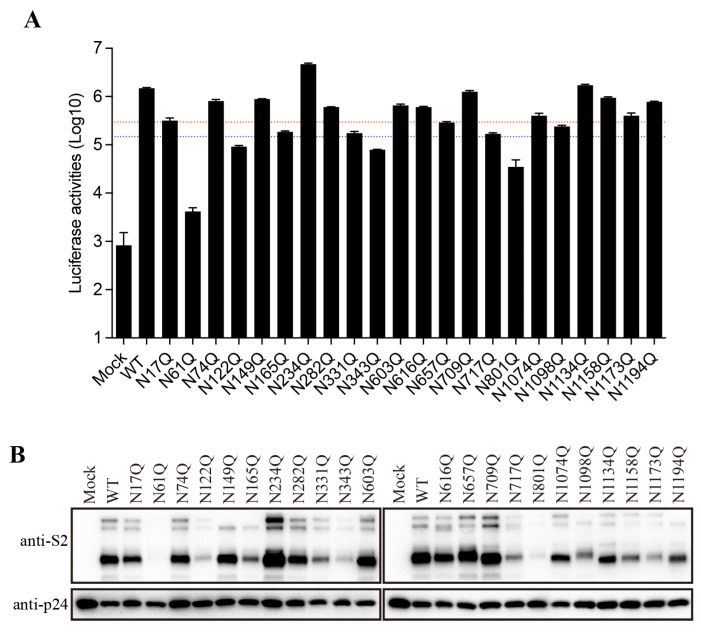
The role of SARS-CoV-2 S protein N-glycans in viral entry. (**A**) The transduction efficiency of wild-type or N-glycosylation-mutant S-pseudotyped lentiviral particles. The red and blue dashed lines represent 1/5 and 1/10 of the wild-type transduction efficiency, respectively. (**B**) Western blot analysis of wild-type or N-glycosylation-mutant S-pseudotyped lentiviral particles.

**Figure 5 viruses-16-00223-f005:**
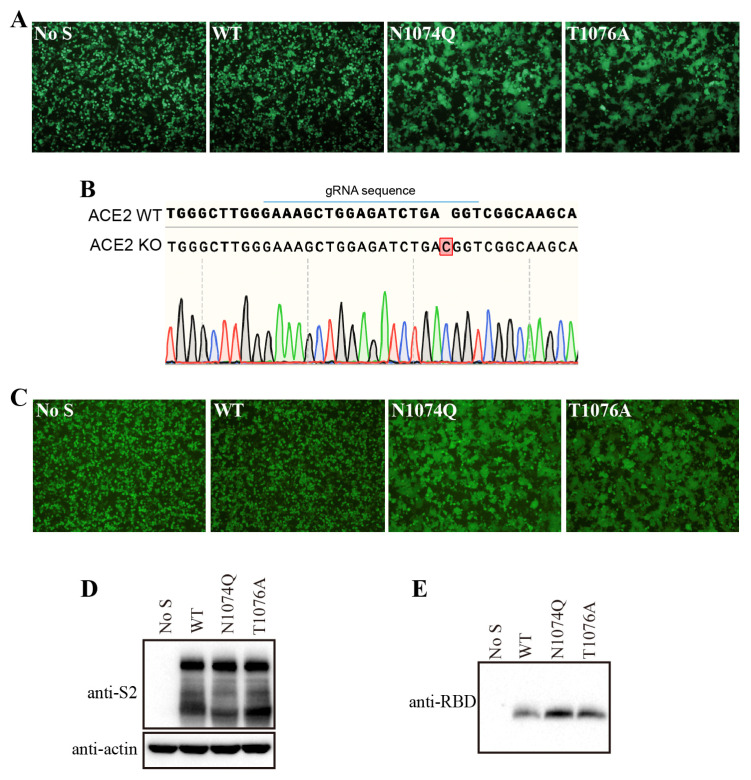
Removal of N-glycans from the SARS-CoV-2 S protein destabilized the S protein. (**A**) HEK293T cells were co-transfected with plasmids encoding EGFP and plasmids encoding wild-type SARS-CoV-2 S protein, the N1074Q mutant, or the T1076A mutant. After 40 h of transfection, the cells were visualized using a fluorescence microscope. (**B**) Endogenous ACE2 knockout in HEK293T cells was confirmed through DNA sequencing. A single-nucleotide insertion in the ACE2 gene caused a frameshift and led to the premature termination of ACE2 translation. (**C**) The endogenous-ACE2-knockout HEK293T cells were co-transfected with plasmids encoding EGFP and plasmids encoding wild-type SARS-CoV-2 S protein, the N1074Q mutant, or the T1076A mutant. After 40 h of transfection, the cells were visualized using a fluorescence microscope. (**D**,**E**) HEK293T cells were transfected with plasmids encoding wild-type SARS-CoV-2 S protein, the N1074Q mutant, or the T1076A mutant. After 40 h of transfection, the S protein in cell lysates (**D**) and supernatants (**E**) was analyzed via Western blot with antibodies against the S2 subunit and the RBD subunit of the SARS-CoV-2 S protein, respectively.

**Figure 6 viruses-16-00223-f006:**
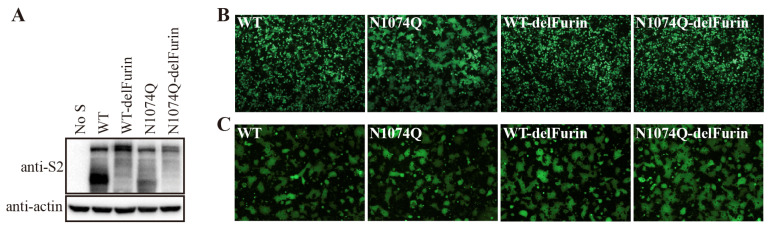
Removal of the furin cleavage site compensated for the instability of the N-glycosylation mutant N1074Q. (**A**) HEK293T cells were transfected with plasmids encoding wild-type SARS-CoV-2 S protein, wild-type S protein with a mutant furin cleavage site (WT-delFurin), mutant N1074Q, or N1074Q with a mutant furin cleavage site (N1074Q-delFurin). After 40 h of transfection, the cells were lysed and analyzed via Western blot with antibodies against the SARS-CoV-2 S2 subunit. β-actin was used as a loading control. (**B**) HEK293T cells were co-transfected with plasmids encoding GFP and plasmids encoding wild-type SARS-CoV-2 S protein, the WT-delFurin mutant, the N1074Q mutant, or the N1074Q-delFurin mutant. After 40 h of transfection, the cells were visualized using a fluorescence microscope. (**C**) A cell–cell fusion assay was performed to assess the fusogenicity of wild-type SARS-CoV-2 S protein, the WT-delFurin mutant, the N1074Q mutant, and the N1074Q-delFurin mutant. HEK293T cells were co-transfected with plasmids encoding wild-type or mutant S proteins and plasmids encoding GFP. After 30 h of transfection, the cells were trypsinized and co-cultured with HEK293/hACE2 cells for an additional 4 h and then imaged using a fluorescence microscope.

**Figure 7 viruses-16-00223-f007:**
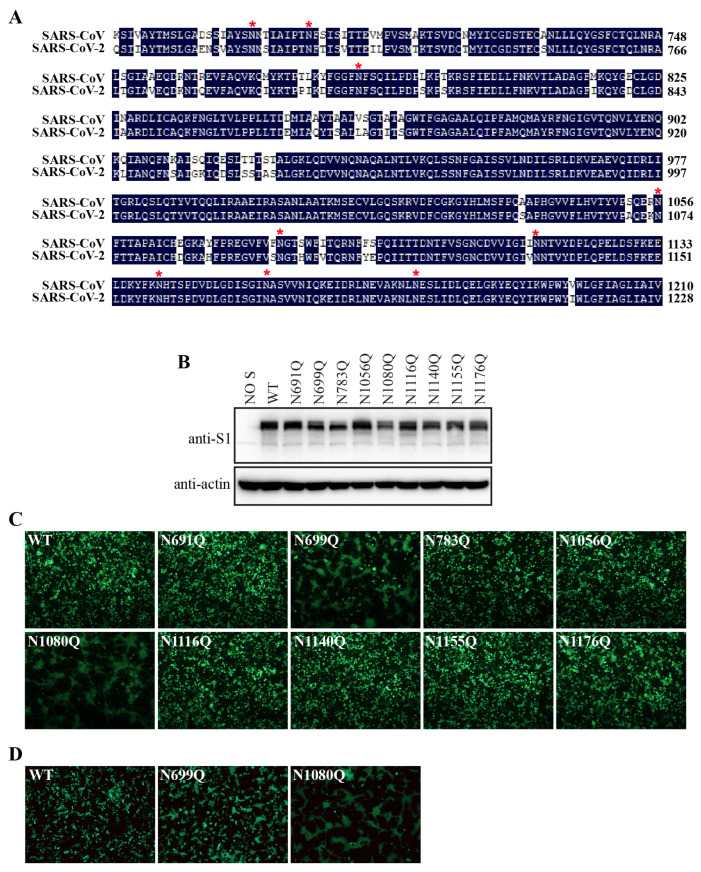
Removal of N-glycans from the S2 subunit of SARS-CoV S protein destabilized the S protein. (**A**) Amino-acid sequence alignment of S2 subunits from SARS-CoV and SARS-CoV-2. Red asterisks indicate N-linked glycosylation sites. (**B**) HEK293T cells were transfected with an empty vector or with plasmids encoding the wild-type SARS-CoV S protein or one of the various glycosylation mutants. After 40 h of transfection, the cells were lysed and analyzed with polyclonal rabbit anti-SARS S1 antibodies T62. β-actin was used as a loading control. (**C**) HEK293T cells were co-transfected with plasmids encoding EGFP and plasmids encoding wild-type SARS-CoV S protein or one of the various glycosylation mutants. After 40 h of transfection, the cells were visualized using a fluorescence microscope. (**D**) HEK293T/ACE2-KO cells were co-transfected with plasmids encoding EGFP and plasmids encoding wild-type SARS-CoV S protein, the N699Q mutant, or the N1080Q mutant. After 40 h of transfection, the cells were visualized using a fluorescence microscope.

**Figure 8 viruses-16-00223-f008:**
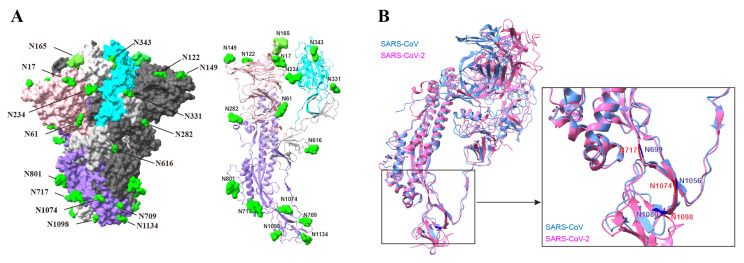
N-linked glycosylation sites on the structure of the SARS-CoV-2 S protein. (**A**) Pre-fusion structure of the trimeric SARS-CoV-2 S proteins with all RBDs in the closed state (PDB 7qus). Each monomer is shown in a different shade of gray. The NTD, RBD, and S2 subunit of one S monomer are shown in pink, cyan, and purple, respectively. One S monomer structure was shown as a cartoon. (**B**) Structural comparison of the SARS-CoV-2 and SARS-CoV S monomers. SARS-CoV S protein PDB accession number: 5x58 (blue); SARS-CoV-2 S protein PDB accession number: 7qus (pink).

## Data Availability

The data presented in this study are available on request from the corresponding authors.

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
