# Peer review of "Stabilization of the Metastable Pre-Fusion Conformation of the SARS-CoV-2 Spike Glycoprotein through N-Linked Glycosylation of the S2 Subunit"

_viruses, 2024, doi:10.3390/v16020223_

Round 1

Reviewer 1 Report

Comments and Suggestions for Authors

The paper is of interest, the conclusions are fair and justified by the data presented and methods adopted. I have only a few remarks/requests of clarification:

1. Although the alfa-glycosidase inhibitors used in this work are generally considered safe, I wonder whether the Authors have evidence of a total lack of toxicity in cell cultures treated for 40 hrs with 0.1 mg of castanospermine or NB-DNJ. A clear statement about this should be added.

2. The alfa-glycosidase inhibitors are known to act on N-glycan editing ,and have  been reported as inhibitors of   complex (mostly sialydated or fucosylated glycans), leaving intact the simpler, mannosylated chains ( see Pili R et al. in Cancer Research 1995, 55,2920). In contrast, in the experiments of site-directed mutagenesis, where asparagine is replaced by glutamine, no N-glycan at all is present. Intriguingly,  both approaches lead essentially to the same result , i.e. lack or great diminution of S protein cleavage. Would it mean that the nascent, non-edited N-glycan chain hasn't any relevance or role for S protein stability and virus infection? Are the N-glycan sites which have been shown to be important for S protein stability those with more complex, edited glycans? The Authors may wish to clarify to the readers this mechanistic aspect.

3.Although the manuscript is readable and its content clearly understandable, there are several typos and grammar errors all throughout it ( just to show here some of them : line 44: evolve; line379: became; line 451: trimer). Overall, the  English usage needs a careful editorial check

Comments on the Quality of English Language

see my third comment in the Suggestions to the Authors above

Reviewer 2 Report

Comments and Suggestions for Authors

The authors have experimentally analyzed the effects of N-glycans on the SARS-CoV-2 spike protein, focusing on its expression, stability, membrane fusion, and viral entry. Their mutagenesis study indicates that 9 out of 22 N-linked glycosylation sites are critical for proper folding and maturation of the spike protein, which subsequently affects cell-cell fusion and viral entry. This study could potentially offer significant insights into the roles of glycans in the spike protein of SARS-CoV-2. However, I have a few concerns that need to be addressed for further consideration.

My primary observation is that the discussion on the roles of individual glycans lacks the context of their spatial locations within the spike protein's three-dimensional structure. Figure 8 presents a structural visualization, but it is not easy to interpret. The roles of other glycosylation sites should also be examined within the structural context, as several studies have done so. I recommend the authors integrate their findings with the broader structural context of the spike protein.

Regarding Figure 1A, the authors suggest that two close bands correspond to full-length (FL) spike proteins, with the higher faint band possibly representing the fully glycosylated form. They further postulate that treatment with α-glycosidase inhibitors had little effect on FL S proteins but significantly decreased the levels of cleaved S proteins compared to the DMSO-treated group. However, it seems to me that after post-treatment, the lower band becomes fainter, which suggests that the inhibitors do affect the levels of FL S proteins.

In Figure 1B, it is claimed that the size and frequency of syncytium formation were significantly reduced in HEK293T cells expressing the SARS-CoV-2 spike protein after pre-treatment with castanospermine or NB-DNJ. Yet, the visual differences between the control and the NB-DNJ-treated cells are not immediately apparent. Clarification on this observation would be helpful.

Round 2

Reviewer 2 Report

Comments and Suggestions for Authors

The authors addressed all of my previous comments properly.